# Factors driving choices between types and brands of influenza vaccines in general practice in Austria, Italy, Spain and the UK

Anke L. Stuurman[1,2]*, Sara Ciampini[3], Alfredo Vannacci[4], Antonino Bella[5], Caterina Rizzo[6], Cintia Muñoz-Quiles[7], Elisabetta Pandolfi[3], Harshana Liyanage[8], Mendel Haag[9], Monika Redlberger-Fritz[10], Roberto Bonaiuti[4], Philippe Beutels[2]

1 P95 Epidemiology & Pharmacovigilance, Leuven, Belgium, 2 Centre for Health Economics Research & Modelling Infectious Diseases, Vaccine & Infectious Disease Institute, University of Antwerp, Antwerp, Belgium, 3 Multifactorial Disease and Complex Phenotype Research Area, Bambino Gesù Children's Hospital, IRCCS, Rome, Italy, 4 Department of Neurosciences, Psychology, Drug Research and Child Health, University of Florence, Firenze, Italy, 5 Department of Infectious Diseases, National Institute of Health, Rome, Italy, 6 Functional Area of Clinical Patways and Epidemiology, Bambino Gesù Children's Hospital, IRCCS, Rome, Italy, 7 Vaccines Research Unit, Fundación para el Fomento de la Investigación Sanitaria y Biomédica de la Comunitat Valenciana, FISABIO Public Health, Valencia, Spain, 8 Nuffield Department of Primary Care Health Sciences, University of Oxford, Oxford, United Kingdom, 9 Epidemiology, Medical Affairs, Seqirus, Amsterdam, The Netherlands, 10 Center for Virology, Medical University of Vienna, Vienna, Austria

* anke.stuurman@p-95.com

**Data Availability Statement:** All relevant data are within the manuscript and S1 and S2 Files.

## Abstract

Influenza vaccine effectiveness (IVE) assessment is increasingly stratified by vaccine type or brand, such as done by the European network of DRIVE. In 2019/2020, eleven influenza vaccines were licensed in Europe. If more than one vaccine type is recommended or if more than one vaccine brand is available for a specific risk group, it is not clear which factors affect the choice of a specific vaccine (type or brand) by a health practitioner for individual patients. This is important for IVE assessment. A survey tailored to the 2019/20 local vaccine recommendations was conducted among GPs in four European countries (Austria, Italy, Spain, UK) to understand how influenza vaccine is offered to recommended risk groups and, if GPs have a choice between 2 or more vaccines, what factors influence their vaccine choice for patients. Overall, 360 GPs participated. In Austria, Italy and Spain GPs indicated that influenza vaccines are commonly offered when patients present for consultation, whereas in the UK all GPs indicated that all relevant patients are contacted by letter. In Austria and Italy, roughly 80% of GPs had only one vaccine type available for patients <65y. The use of any specific vaccine type in this age group is mostly determined by the availability of specific vaccine type(s) at the clinic. GPs frequently reported availability of more than one vaccine type for patients ≥65y in Austria (45%), Italy (70%) and Spain (79%). In this group, patient characteristics played a role in choice of vaccine, notably older age and presence of (multiple) comorbidities. Knowing that a non-patient related factor usually determines the vaccine type a patient receives in settings where more than one vaccine type is recommended for risk groups <65y, simplifies IVE assessment in this age group. However, patient characteristics need careful consideration when assessing IVE in those ≥65y.

**Funding:** EU/EFPIAIMI-2 Joint Undertaking (DRIVE, grant n ̊777363). The funder provided support in the form of salaries for authors AS, SC, AV, AB, CR, CMQ, EP, HL, MH, and RB, but did not have any additional role in the study design, data collection and analysis, decision to publish, or preparation of the manuscript. The specific roles of these authors are articulated in the 'author contributions' section.

**Competing interests:** Phillipe Beutels reports grants from Innovative Medicines Initiative of the European Commission and attended meetings of an advisory board on economic evaluations of vaccines convened by Pfizer in 2019, outside the submitted work. Anke Stuurman is employed by P95 Epidemiology & Pharmacovigilance. Mendel Haag is employed by Seqirus BV. This does not alter our adherence to PLOS ONE policies on sharing data and materials.

## Introduction

Influenza vaccine effectiveness (IVE) assessment is increasingly stratified by vaccine type [1–4] or even vaccine brand, such is the IVE assessment performed by the DRIVE network in the EU [5]. DRIVE is an IMI-funded public-private partnership that aims to estimate brand-specific IVE every season to fulfil the requirements of the European Medicines Agency [6], and (type-specific) IVE to inform public health decision-making. Seasonal influenza vaccination is widely recommended across Europe, although the exact recommendations vary across countries [7]. Vaccine uptake varies, for example vaccine coverage for older age groups ranged from 2% to 72.8% across 19 EU/EEA Members States in the 2016–17 season [7].

In 2019/2020, eleven influenza vaccines were licensed in Europe. The majority were inactivated vaccines, which differ in valency (trivalent or quadrivalent), production process (egg-based or cell-based) and may contain a higher dose of antigen or include an adjuvant to enhance the immune response. A quadrivalent live attenuated vaccine was also available. In some countries, a single vaccine type was available for use in all risk groups, whereas in others, several vaccine types were available and the local influenza vaccination recommendations may describe which vaccine type or types to use in which risk group. Previous work on seasonal influenza vaccine procurement has shown that one or multiple brands of one vaccine type may be procured [8].

In the event that more than one vaccine type is recommended or where more than one specific vaccine brand is available for a specific risk group, it is not clear which factors affect the choice of a specific vaccine (type or brand) by a health practitioner for an individual patient. These could include factors such as patient preference (e.g. for nasal vs. injectable vaccine), vaccine price or reimbursement, patient characteristics (e.g. older age, presence of multiple or more severe comorbidities), or be determined by vaccine availability (reflecting procurement or stock at the GP practice or pharmacy). Specifically, understanding whether patient characteristics impact the choice of vaccine type within a given risk group is important for IVE assessment, as this could result in differences in baseline characteristics between vaccinated persons and hence introduce bias in the IVE if not properly accounted for.

To this end, a survey among GPs in four countries was conducted. Specifically, the aims were to understand (1) how influenza vaccine is offered to patients in risk groups and the extent to which patients' risk perception or health conscious behaviour influences vaccination acceptance; (2) whether GPs have a choice between 2 or more vaccine types for patients in a specific risk group, and if so what factors influence their vaccine choice; and (3) whether more than one brand of a specific vaccine type is available, and if GPs consider different brands of the same vaccine type to be clinically equivalent.

## Methods

### Survey

A survey consisting of three parts, reflecting each of the objectives described above, was developed. The target respondents of the survey were GPs in Austria, Italy, Spain (Valencia region), and the UK (England). The part of the survey on vaccine types, was tailored to local influenza vaccine type recommendations for 2019/2020 (Table 1) [9–12].

The four countries were chosen because sites participating in DRIVE are located in these countries and because there was at least one risk group for which more than one vaccine could be used in the 2019/2020 season. In Austria, Italy and the UK more than one vaccine type was recommended for at least one risk group in the national influenza vaccine recommendations. Spain-Valencia was selected before the publication of the 2019/2020 regional recommendations

**Table 1. Risk groups covered by the survey and influenza vaccine types recommended for use in these risk groups, by country, 2019/2020.**

|  | Austria | Italy [c] | Spain-Valencia | UK-England |
|---|---|---|---|---|
| **Children 6m-17y** | QIVe or QIVc (≥9y-17y)[a] | *Risk groups* QIVe or QIVc (≥10y-17y) | - | - |
| **Adults 18-64y** | QIVe, QIVc | *Risk groups* QIVe, QIVc | - | *Risk groups* QIVe, QIVc |
| **Adults ≥65y** | aTIV, QIVe[b], QIVc[b] | aTIV, QIVe, QIVc (aTIV is preferentially recommended in those aged ≥75y) | aTIV | aTIV, QIVc, TIV-HD |
| **Any person with comorbidities or immunosuppression** | aTIV, QIVe, QIVc | - | - | - |

aTIV: adjuvanted trivalent influenza vaccine; m; months; QIVc: cell-based inactivated quadrivalent influenza vaccine; QIVe: egg-based inactivated quadrivalent influenza vaccine; TIV-HD: high-dose trivalent influenza vaccine; y: years.

[a] Children: live attenuated influenza vaccine also recommended for children >2y, however not available in Austria in 2019/2020.

[b] Adults ≥65y: in the event of intensive circulation of B strain only covered by quadrivalent vaccines.

[c] TIV also recommended however this vaccine was not procured by Italian regions in 2019/2020.

in October 2019 and inclusion was based on the 2018/2019 recommendations, in which two vaccine types were recommended in those aged 65 years and above. Despite no longer recommending two types of vaccine for the same age group in 2019/2020, we decided to keep Spain-Valencia because it was observed that in practice two vaccine types were still being used. The second part of the survey focuses on vaccine choice in these risk groups (Table 1).

The survey was translated from English into German, Italian and Spanish, as appropriate, and piloted with one or two GPs per country. Their feedback was incorporated into the final version.

All GPs were approached through the DRIVE network. In Austria and Italy, all GPs participating in the national sentinel influenza surveillance were invited to participate in the survey (n = 92 in Austria, n = 1179 in Italy); in Austria invitations were sent by email, in Italy the survey was posted on the restricted section of the Italian surveillance website where sentinel GPs log-in weekly. In Spain, the survey was shared with an unknown number of GPs by two GP associations in Valencia. In the UK, all GPs participating in DRIVE were approached (n = 12).

The survey was prepared in online-format using SurveyMonkey® [13]. Multiple responses from the same device were disabled. In Italy and Spain, the survey was completed online. In Austria, the GPs could choose to complete the survey online or on paper. In the UK, the GPs were phoned to increase the response rate given the limited number of GPs. All answers obtained on paper or by phone were transferred to the online survey. The surveys were completed between December 2019 and January 2020 (Austria: 11/12 to 19/01; Italy: 10/12 to 16/01; Spain: 10/12-5/01; the UK: 03/01 to 21/01;).

Several response options were used in the survey: open-ended questions, closed-ended questions with a single answer option (including Likert scales) and closed-ended questions with multiple answer options. The surveys are available in S1 File.

## Statistical methods

When possible, response rates were calculated. Response was defined as answering at least the first question beyond individual GP's demographics. Descriptive analyses of the close-ended questions (counts, proportions) were performed by country and by applicable risk group in each country. For the open-ended questions, similar responses were grouped and the number of responses corresponding to each group was quantified. The groups were defined a posteriori.

SurveyMonkey's question summaries [13] and Excel were used for the analyses.

### Ethics

This study was approved by the ethics committee of the University of Antwerp / University Hospital Antwerp on December 2, 2019 (B300201942312).

## Results

Overall, 360 GPs participated in the survey. The response rate was 59.8% (55 out of 92) in Austria, approximately 14.2% in Italy (167 out of 1179), and 100% in the UK (12 out of 12). There were 126 respondents in Spain. The response rate for Spain could not be calculated as the number of GPs that received the survey is not known. Characteristics of GPs and their practice are described in Table 2. Raw data is presented in S2 File.

### Offer and acceptance of influenza vaccine

National or regional vaccine recommendations indicate for which risk groups influenza vaccine is recommended. Different approaches to offering influenza vaccine to these risk groups exist, and often multiple approaches are in place in one practice (Table 3). GPs in Austria, Italy and Spain indicated that influenza vaccines are most commonly offered when a patient presents for a consultation at the time of the vaccination campaign or during the influenza season. In the UK, however, all surveyed GPs indicated that all relevant patients are contacted by letter and offered influenza vaccine; this approach was also reported by a third of the GPs in Italy. Other approaches reported by GPs in Austria, Italy and Spain included posters in the waiting

**Table 2. Characteristics of GPs participating in the survey.**

|  | Austria | Italy | Spain-Valencia | UK—England |
|---|---|---|---|---|
|  | n (%) | n (%) | n (%) | n (%) |
| **Total GPs** | 55 | 167 | 126 | 12 |
| **Years of practice** |  |  |  |  |
| *0–9* | 8 (14.5) | 10 (6.0) | 37 (29.4) | 0 (0) |
| *10–19* | 16 (29.1) | 14 (8.4) | 23 (18.3) | 6 (50.0) |
| *20–29* | 13 (23.6) | 32 (19.2) | 30 (23.8) | 4 (33.3) |
| *30–39* | 17 (30.9) | 93 (55.7) | 35 (27.8) | 2 (16.7) |
| *≥40* | 1 (1.8) | 18 (10.8) | 1 (0.8) | 0 (0) |
| **Populations for which influenza vaccination is provided at practice [a]** |  |  |  |  |
| *Children* | 44 (80.0) | 67 (40.1) | 45 (35.7) | 12 (100) |
| *Adults <65y* | 44 (80.0) | 128 (76.6) | 38 (30.2) | 12 (100) |
| *Pregnant women* | 30 (54.5) | 94 (56.3) | 115 (91.3) | 12 (100) |
| *Adults ≥65y* | 43 (78.2) | 134 (80.2) | 123 (97.6) | 12 (100) |
| *Institutionalized* | 32 (58.2) | 94 (56.3) | 120 (95.2) | 12 (100) |
| **Region** | See footnote [b] | See footnote [c] | Valencia (126) | England, see footnote [d] |

[a] Multiple answers possible, therefore may not tally to 100%.

[b] Region (number of GPs participating in survey): Berkshire (1), East Sussex (1), Kent (1), London (2), Oxfordshire (5), Shire (1), Somerset (1).

[c] Region (number of GPs participating in survey): Abruzzo (6), Basilicata (2), Calabria (1), Campania (5), Emilia Romagna (19), Friuli Venezia Giulia (3), Lazio (15), Liguria (5), Lombardia (25), Marche (5), Piemonte (14), Puglia (24), Sardegna (1), Sicilia (16), Toscana (15), Trentino Alto Adige (3), Trento (1), Valle d'Aosta (2), Veneto (5).

[d] Burgenland (2), Kärnten (1), Niederösterreich (10), Oberösterreich (8), Salzburg (5), Steiermark (9), Tirol (5), Vorarlberg (1), Wien (10), not reported (4).

room. In Spain, several GPs (n = 7 (5.6% of GPs who were asked to comment on how influenza vaccination is offered)) commented on their lack of proactivity in offering influenza vaccine.

GPs also indicated the extent to which they agreed with the statements that patients are more likely to accept vaccination (1) if patients perceive themselves to be at higher risk and (2) if patients seem more health conscious (Table 3). In Austria, Italy and Spain, over 90% of GPs 'agreed fully' or 'mostly agreed' with both statements. However, among GPs in the UK, half 'mostly agreed' with the first statement and half 'mostly disagreed', and two thirds 'mostly agreed' with the second statement, and one third 'mostly disagreed'.

GPs were also asked to comment (open-ended response format) on other factors that influence vaccine acceptance. GPs commented that patients who have experienced an influenza infection (themselves or in the family) are more inclined to accept vaccination (Austria n = 6 (11.1% of GPs who were asked to comment on factors that make their patients more likely to accept vaccination), Italy n = 6 (3.6%) and Spain n = 5 (4.0%)), as are those who perceive the benefits of vaccination (Italy n = 22 (13.2%)) and those who perceive vaccination as contributing to the health of the community at large and the protection of others (Austria n = 1 (1.9%), Italy n = 1 (0.6%), Spain n = 6 (4.8%)). GPs reported that a relationship of trust between the patient and the health care professional (Italy n = 13 (7.8%), Spain n = 17 (13.6%)) and counselling regarding

**Table 3. Description of how seasonal influenza vaccine is offered to individuals in risk groups that are part of the vaccine recommendations, in 2019/2020, by country.**

| | Austria | Italy | Spain-Valencia | UK-England |
|---|---|---|---|---|
| | n (%) | n (%) | n (%) | |
| **How is seasonal influenza vaccine offered to individuals in risk groups that are part of the vaccine recommendations?** [a] | (N = 55) | (N = 167) | (N = 126) | (N = 12) |
| *All are contacted* | 2 (3.6) | 55 (32.9) | 20 (15.9) | 12 (100) |
| *A subset is contacted* | 1 (1.8) | 16 (9.6) | 4 (3.2) | 0 (0) |
| *All are proactively offered influenza vaccination if they present for a consultation during the vaccination campaign or influenza season* | 45 (81.8) | 85 (50.9) | 105 (83.3) | 0 (0) |
| *A subset is proactively offered influenza vaccination if they present for a consultation during the vaccination campaign or influenza season* | 11 (20.0) | 12 (7.2) | 26 (20.6) | 0 (0) |
| *Only upon request from the patient* | 9 (16.4) | 18 (10.8) | 6 (4.8) | 0 (0) |
| *Other* | 12 (21.8) | 33 (19.8) | 25 (19.8) | 0 (0) |
| **Patients who perceive themselves to be at higher risk are more likely to accept vaccination** | (N = 55) | (N = 167) | (N = 126) | (N = 12) |
| *Fully agree* | 28 (50.9) | 79 (47.3) | 54 (42.9) | 1 (8.3) |
| *Mostly agree* | 23 (41.8) | 76 (45.5) | 69 (54.8) | 5 (41.7) |
| *Mostly disagree* | 4 (7.3) | 9 (5.4) | 4 (3.2) | 6 (50.0) |
| *Fully disagree* | 0 (0) | 3 (1.8) | 0 (0) | 0 (0) |
| **Patients who are more health conscious are more likely to accept vaccination** | (N = 54) | (N = 167) | (N = 125) | (N = 12) |
| *Fully agree* | 27 (50.0) | 100 (59.9) | 43 (34.4) | 0 (0) |
| *Mostly agree* | 23 (42.6) | 62 (37.1) | 74 (59.2) | 8 (66.7) |
| *Mostly disagree* | 3 (5.6) | 3 (1.8) | 8 (6.4) | 4 (33.3) |
| *Fully disagree* | 1 (1.9) | 2 (1.2) | 1 (0.8) | 0 (0) |

n: Number of GPs that chose a particular answer; N: number of GPs that answered the question.

[a] Multiple answers possible, therefore may not tally to 100%.

vaccination (Austria n = 3 (5.6%), Italy n = 28 (16.8%), Spain n = 9 (7.2%)) were key to vaccination acceptance. Prior experience with the influenza vaccine influenced both the probabilities of vaccine acceptance and rejection (Italy n = 6 (3.6%), Spain n = 9 (7.2%)). A minority reported fear of side effects as a deterrent to vaccination (Italy n = 4 (2.4%)). GPs also emphasized the importance of external factors outside of the patient, namely the role of the media in both increasing vaccine uptake (Austria n = 4 (7.4%), Italy n = 5 (3.0%)), and decreasing uptake ("fake news") (Italy n = 3 (1.8%)). Finally, the accessibility of vaccination (e.g. in terms of price, time required, possibility of receiving vaccine at the practice) was reported to be important in the acceptance of vaccine (Austria n = 3 (5.6%), Italy n = 5 (3.0%)).

## Choice of vaccine type

In Austria, availability of vaccine types was studied in children 9-17y, adults 18-64y, patients with a chronic condition or immunosuppression and adults ≥65y (Table 4). Roughly 80% of GPs had only one vaccine type available for children and for healthy adults 18-64y, typically egg-based QIV (Fig 1). Almost half of GPs had more than one vaccine type available for patients with chronic conditions or immunosuppression (43.4%) or adults ≥65y (45.2%).

Even when more than one vaccine type was available in the season, the use of any specific type at a given time was frequently directed by vaccine availability at the pharmacy (range from 42.9% to 65.2%). Patient characteristics did not usually influence the choice of vaccine type for children and adults <65y, but did for patients with chronic conditions or immunosuppression (69.6%) and adults ≥65y (73.7%).

**Table 4. Vaccine type availability and factors that influence choice of vaccine type by the GP in 2019/2020, by risk group, Austria.**

| AUSTRIA | 9-17y | 18-64y | Chronic conditions or immunosuppression | ≥65y |
|---|---|---|---|---|
| | n (%) | n (%) | n (%) | n (%) |
| **Vaccine types available** | (N = 51) | (N = 45) | (N = 53) | (N = 42) |
| *1* | 40 (78.4) | 38 (84.4) | 30 (56.6) | 23 (54.8) |
| *>1* | 11 (21.6) | 7 (15.6) | 23 (43.4) | 19 (45.2) |
| **The following factors influence choice of vaccine type by the GP[a]** | (N = 11) | (N = 7) | (N = 23) | (N = 19) |
| *Patient preference* | 2 (18.2) | 2 (28.6) | 4 (17.4) | 5 (26.3) |
| *Willingness to pay* | 3 (27.3) | 1 (14.3) | 3 (13.0) | 5 (26.3) |
| *Price* | 1 (9.1) | 1 (14.3) | 2 (8.7) | 0 (0) |
| *Directed by vaccine availability* | 6 (54.5) | 3 (42.9) | 15 (65.2) | 12 (63.2) |
| *Patient characteristics* | 2 (18.2) | 1 (14.3) | 16 (69.6) | 14 (73.7) |
| **The following patient characteristics influence choice of vaccine type by the GP[b]** | (N = 2) | (N = 1) | (N = 14) | (N = 14) |
| *Younger age* | 1 (50.0) | 0 (0) | 4 (28.6) | 3 (21.4) |
| *Older age* | 2 (100) | 0 (0) | 14 (100.0) | 14 (100) |
| *Presence of multiple comorbidities* | 1 (50.0) | 0 (0) | 9 (64.3) | 6 (42.9) |
| *Presence of specific comorbidities* | 0 (0) | 1 (100) | 2 (14.3) | 1 (7.1) |
| *Higher severity of comorbidities* | 0 (0) | 0 (0) | 5 (35.7) | 5 (35.7) |
| *Frailty* | 0 (0) | 0 (0) | 4 (28.6) | 3 (21.4) |
| *Profession* | - | 1 (100) | - | - |
| *Other* | 0 (0) | 0 (0) | 0 (0) | 0 (0) |

aTIV: adjuvanted trivalent influenza vaccine; n: number of GPs that chose a particular answer; N: number of GPs that answered the question; QIVe: egg-based quadrivalent influenza vaccine; QIVc: cell-based quadrivalent influenza vaccine.

[a] This question was only posed to GPs that answered ">1 type" to the previous question.

[b] This question was only posed to GPs that answered "patient characteristics" to the previous question.

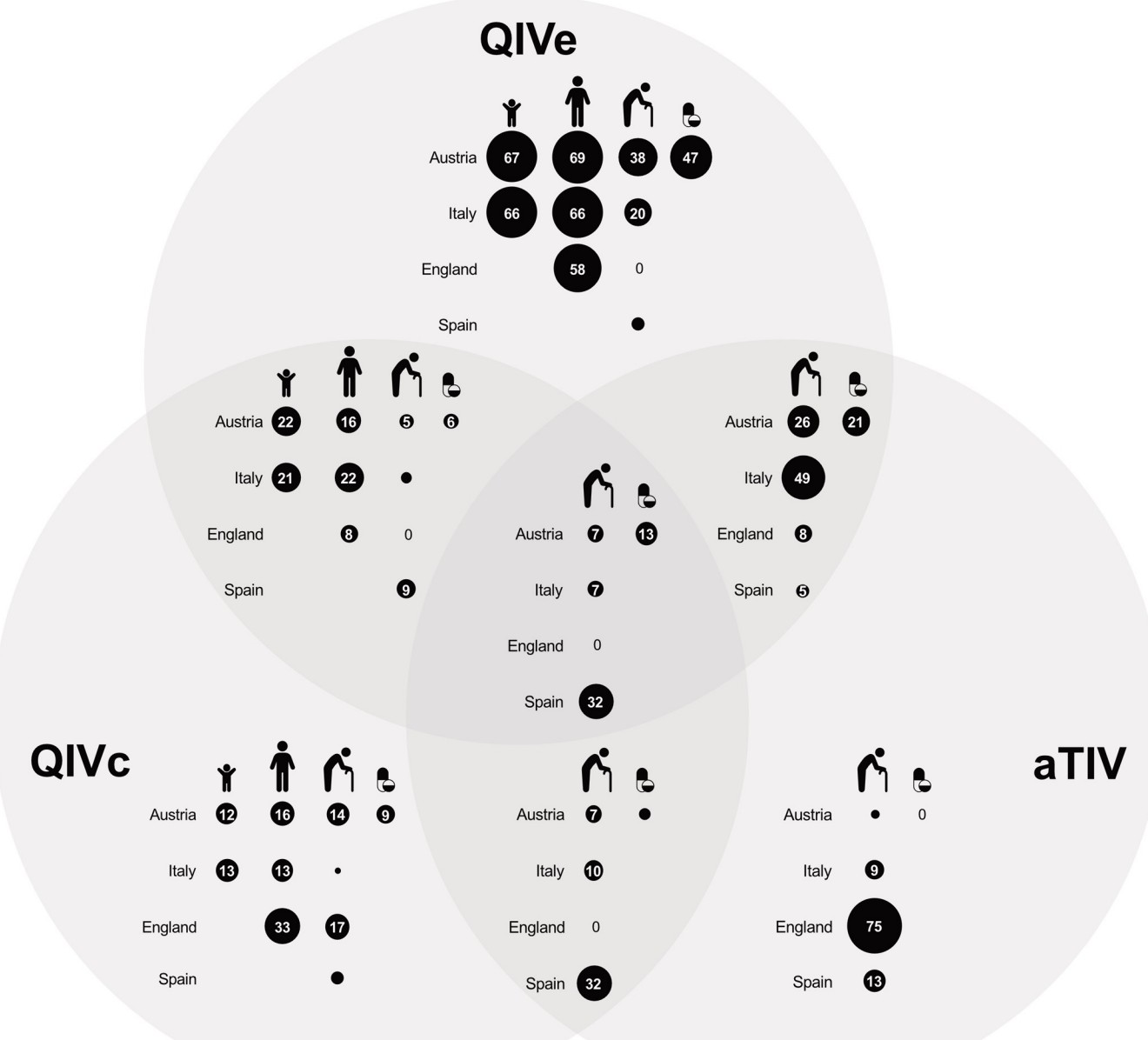

**Fig 1. Vaccine type availability by risk group and country in 2019/20, as indicated by GPs (%).** The risk groups are 1) children (healthy children 9-17y in Austria, children 10-17y with chronic conditions in Italy), 2) adults 18-64y (healthy adults in Austria, adults with chronic conditions in Italy and England), 3) adults ≥65y, and 4) persons with chronic conditions or immunosuppression 18-64y in Austria. Empty circles represent values lower than 5.

Multiple GPs (n = 7) reported bottlenecks in delivery of vaccines to pharmacies in recent years, which resulted in a switch from the preferred vaccine (or from adjuvanted TIV to a quadrivalent vaccine in the absence of intensive circulation of the B strain covered only by quadrivalent vaccines). The most influential patient characteristic in both groups was older age (100%), and GPs indicated they more frequently chose adjuvanted TIV for older patients. Among patients with chronic conditions or immunosuppression, another important characteristic was the presence of multiple comorbidities (64.3%).

In Italy, availability of vaccine types was studied in children 10-17y with chronic conditions, adults 18-64y with chronic conditions, and adults ≥65y (Table 5). Nearly 80% of GPs had only one vaccine type available for children and for adults 18-64y, typically egg-based QIV (Fig 1). Even when more than one vaccine type was available in the season, the use of any specific vaccine type at a given time was frequently driven by vaccine availability in these two age groups (64.3% and 69.6%), although patient characteristics also played a role for half of the GPs for the adult population. In particular, GPs indicated older age and presence of multiple comorbidities (both 72.7%) influenced their choice of vaccine type among adults 18-64y. For the ≥65y age group, the majority of GPs had more than one vaccine available (69.8%), and typically included adjuvanted TIV and egg-based QIV. In Italy, adjuvanted TIV was preferentially recommended for adults ≥75y at the time the survey was conducted. The majority of GPs (85.5%) indicated adjuvanted TIV was usually or always administered to those ≥75y,

**Table 5. Vaccine type availability and factors that influence choice of vaccine type by the GP in 2019/2020, by risk group, Italy.**

| ITALY | 10-17y with chronic conditions | 18-64y with chronic conditions | ≥65y | 65-74y | ≥75y |
|---|---|---|---|---|---|
| | n (%) | n (%) | n (%) | n (%) | n (%) |
| **Vaccine types available** | (N = 139) | (N = 116) | (N = 116) | | |
| *1* | 110 (79.1) | 91 (78.4) | 35 (31.0) | - | - |
| *>1* | 29 (20.9) | 25 (21.6) | 81 (69.8) | - | - |
| *>1 excluding aTIV* | - | - | 4 (3.4) | - | - |
| *>1 including aTIV* | - | - | 77 (66.4) | - | - |
| ***Frequency with which GP chooses aTIV*** [a] | | | | (N = 77) | (N = 76) |
| *Always* | - | - | - | 8 (10.4) | 21 (27.6) |
| *Usually* | - | - | - | 31 (40.3) | 44 (57.9) |
| *Sometimes* | - | - | - | 23 (29.9) | 5 (6.6) |
| *Not usually* | - | - | - | 13 (16.9) | 6 (7.9) |
| *Never* | - | - | - | 2 (2.6) | 0 (0) |
| **The following factors influence choice of vaccine type by the GP**[b] | | | QIVe vs QIVc (N = 4) | aTIV vs. QIVe and/or QIVc (N = 77) | aTIV vs. QIVe and/or QIVc (N = 76) |
| *Patient preference (N = 28; 24; 4)* [c] | 3 (10.7) | 4 (16.7) | 0 (0) | 14 (18.2) | 15 (19.7) |
| *Directed by vaccine availability (N = 28; 23; 4)* [c] | 18 (64.3) | 16 (69.6) | 4 (100) | 55 (71.4) | 56 (73.7) |
| *Patient characteristics (N = 26; 24; 4)* [c] | 9 (34.6) | 12 (50.0) | 4 (100) | 60 (77.9) | 52 (68.4) |
| **The following patient characteristics influence choice of vaccine type by the GP**[d] | (N = 8) | (N = 11) | (N = 4) | (N = 60) | (N = 52) |
| *Younger age* | 4 (50.0) | 3 (27.3) | 1 (25.0) | 13 (21.7) | 12 (23.1) |
| *Older age* | 7 (87.5) | 8 (72.7) | 1 (25.0) | 50 (83.3) | 42 (80.8) |
| *Presence of multiple comorbidities* | 4 (50.0) | 8 (72.7) | 3 (75.0) | 47 (78.3) | 40 (76.9) |
| *Presence of specific comorbidities* | 1 (12.5) | 0 (0.0) | 0 (0.0) | 15 (25.0) | 11 (21.2) |
| *Higher severity of comorbidities* | 1 (12.5) | 4 (36.4) | 2 (50.0) | 18 (30.0) | 16 (30.8) |
| *Frailty* | 3 (37.5) | 4 (36.4) | 1 (25.0) | 29 (48.3) | 22 (42.3) |
| *Profession* | - | 3 (27.3) | - | | - |
| *Other* | 1 (12.5) | 1 (9.1) | 0 (0.0) | 7 (11.7) | 7 (13.5) |

aTIV: adjuvanted trivalent influenza vaccine; n: number of GPs that chose a particular answer; N: number of GPs that answered the question; QIVe: egg-based quadrivalent influenza vaccine; QIVc: cell-based quadrivalent influenza vaccine.

[a] This question was only posed to GPs that answered ">1 including aTIV" to the first question.

[b] This question was only posed to GPs that answered ">1 type" to the first question.

[c] Number of GPs that answered the question for the risk groups 10-17y with chronic conditions, 18-64y with chronic conditions, and ≥65y, respectively.

[d] This question was only posed to GPs that answered "patient characteristics" to the previous question.

compared to half (50.7%) for those aged 64-75y, which is in line with the national recommendations. Some GPs indicated insufficient doses of adjuvanted TIV were available. Factors influencing the choice of vaccine type were explored separately for adults 65-75y and adults ≥75y among GPs with a choice between adjuvanted TIV and at least one other vaccine type. In both groups vaccine availability (71.4% and 73.7%) and patient characteristics (77.9% and 68.4%) were important determinants of vaccine type use. In particular, older age and presence of multiple comorbidities were found to be important factors that influence choice of vaccine type and GPs commented they were more likely to administer adjuvanted TIV to patients with these characteristics.

In the UK, availability of vaccine types was studied in adults 18-64y with chronic conditions and adults ≥65y. All but one of the GPs (91.7%) indicated they only had one vaccine type available for these two risk groups, either egg-based or cell-based QIV for adults 18-64y or adjuvanted TIV or cell-based QIV for adults ≥65y (Fig 1). Consequently, individual GPs did not have to choose between different vaccine types for their patients. The GP who had more than one vaccine available for the adults ≥65y flagged patient preference as a factor that influenced their choice of vaccine type.

In Spain, availability of vaccine types was studied in adults ≥65y (Table 6). Nearly 80% of GPs had more than one vaccine type available to vaccinate adults ≥65y and typically this included the only vaccine recommended specifically for this age group, adjuvanted TIV (Fig 1). Even when more than one vaccine type was available in the season, use of any specific vaccine type was directed by vaccine availability (70.5%), although patient characteristics also played a role (53.8%). Several GPs reported to typically administer QIVc to the ≥65y

**Table 6. Vaccine type availability and factors that influence choice of vaccine type by the GP in 2019/2020, in the risk group ≥65 years, Spain.**

| SPAIN | |
|---|---|
| | ≥65y |
| | n (%) |
| **Vaccine types available** | (N = 99) |
| *1* | 21 (21.2) |
| *>1* | 78 (78.8) |
| **The following factors influence choice of vaccine type by the GP** [a] | (N = 78) |
| *Patient preference* | 4 (5.1) |
| *Directed by vaccine availability* | 55 (70.5) |
| *Patient characteristics* | 42 (53.8) |
| **The following patient characteristics influence choice of vaccine type by the GP** [b] | (N = 39) |
| *Younger age* | 5 (12.8) |
| *Older age* | 30 (76.9) |
| *Presence of multiple comorbidities* | 26 (66.7) |
| *Presence of specific comorbidities* | 10 (25.6) |
| *Higher severity of comorbidities* | 17 (43.6) |
| *Frailty* | 22 (56.4) |
| *Other* | 9 (23.1) |

aTIV: adjuvanted trivalent influenza vaccine; n: number of GPs that chose a particular answer; N: number of GPs that answered the question; QIVe: egg-based quadrivalent influenza vaccine; QIVc: cell-based quadrivalent influenza vaccine.

[a] This question was only posed to GPs that answered ">1 type" to the previous question.

[b] This question was only posed to GPs that answered "patient characteristics" to the previous question.

population when adjuvanted TIV was not or no longer available. Patient characteristics that influence choice of vaccine type included older age (76.9%), presence of multiple comorbidities (66.7%) and frailty (56.4%), and in such cases GPs commented they were more likely to follow the recommendations for adjuvanted TIV. In addition, multiple GPs reported not recommending vaccination to patients receiving anticoagulant treatment (n = 4).

### Interchangeability of egg-based QIV vaccines

In Italy and Austria, different brands of egg-based QIV were perceived as fully or mostly clinically interchangeable by more than 80% of the GPs (Table 7). In the UK, however, roughly half the GPs mostly agreed and half did mostly not agree.

## Discussion

The main finding from this survey performed among GPs in four EU countries was that the use of any specific vaccine type in the <65y risk group is determined by the availability of specific vaccine type(s) at the clinic level at a given point in time. In addition, for those ≥65y, availability of multiple vaccine types at the clinic level was more common and specifically in this situation the individual characteristics of the vaccinees, notably older age and the presence of multiple comorbidities, play a major role in GPs' choice of vaccine type.

Multiple influenza vaccines types exist. In Europe, conventional trivalent vaccines are being replaced by quadrivalent vaccines and new vaccine types, such as high-dose and cell-based vaccines, are introduced while adjuvanted vaccines have been available longer in some countries, but more recently in others. In the countries studied, local vaccine recommendations recommend one or more specific vaccine types for different risk groups [9–12]. This is also the case in other countries, for example in Europe in Germany [14], Finland [15], Ireland [16], and Spanish regions [17, 18], and beyond in Australia [19] and Canada [20]. It was unknown if in practice GPs have access to more than one vaccine type for each risk group, and if so, how they choose between vaccine types for each individual in a risk group. Only one vaccine type was available for the majority of GPs vaccinating children 9-17y or 10-17y in Austria and Italy, adults 18-64y in Austria, Italy and the UK and adults ≥65y in the UK. More than one of the recommended vaccine types was available for almost half of Austrian GPs vaccinating those with chronic conditions or immunosuppression (of any age) and adults ≥65y, and for the majority of GPs vaccinating adults ≥65y in Italy and Spain.

Vaccine type availability was observed as a determinant of vaccine type use among those aged <65y irrespective of the existence of recommendations for multiple vaccine types; because only a

**Table 7. Egg-based QIV brand availability and perceived clinical interchangeability, by country[a].**

|  | Austria | Italy | UK-England |
|---|---|---|---|
|  | n (%) | n (%) | n (%) |
|  | (N = 55) | (N = 148) | (N = 12) |
| **I have >1 egg-based QIV brand available for my patients** | 19 (34.6) | 15 (10.1) | 1 (8.3) |
| **Perceive egg-based QIV vaccines are clinically interchangeable** | (N = 55) | (N = 146) | (N = 12) |
| *Fully* | 31 (56.4) | 55 (37.7) | 0 (0) |
| *Mostly* | 20 (36.4) | 65 (44.5) | 7 (58.3) |
| *Mostly not* | 2 (3.6) | 15 (10.3) | 5 (41.7) |
| *Not at all* | 2 (3.6) | 9 (6.2) | 0 (0) |

n: number of GPs that chose a particular answer; N: number of GPs that answered the question; QIV: quadrivalent influenza vaccine.

[a] This table was not produced for Spain as a single egg-based QIV brand was available in the Valencia region in 2019/2020.

subset of the recommended vaccine types is made available, or alternatively due to fluctuating vaccine stock. In Austria, Italy and Spain, several GPs indicated that they have encountered bottlenecks in deliveries of vaccines or insufficient stock. Future work may try and explore the underlying reasons for these observations. In Italy, the vaccine viewed by the GP as most appropriate vaccine may not always be administered, as evidenced by surveys among Italian GPs by Levi et al. and Boccalini et al., showing that in the season 2014–2015 more than half of the local health units (60%) did not always allow GPs to choose the type of vaccine for their patients [21], and in the season 2017–2018 only half the GPs had a sufficient number of all available vaccine types they needed [22]. Fluctuations in vaccine availability effectively mean that, even if throughout the season different vaccine types are available to a GP for a risk group, at specific points in the season GPs effectively do not have a choice between vaccine types. Consequently, in this situation vaccine availability is not expected to bias vaccine effectiveness estimation.

The most frequently mentioned patient characteristics to influence choice of vaccine type across all countries (except the UK where typically only one vaccine type was available) were older age followed by the presence of multiple comorbidities. GPs who explained how this affects their vaccine choice indicated they were more likely to choose adjuvanted TIV in the presence of these characteristics. This choice is appropriate as it is in line with the vaccine recommendations in Italy and Valencia which preferentially recommend the adjuvanted vaccine to older age groups due to increased immunogenicity, and is consistent with the findings on more immunogenic vaccines by Boccalini et al. [9, 11, 22–24]. Similarly, in this aforementioned GP survey among Italian GPs it was clear that the majority of GPs were aware that not all influenza vaccines are the same and that some vaccines are preferable to others for some population groups [22]. Knowing that specific vaccine types are sometimes channelled to older and more frail subjects is relevant in an era where vaccine diversity is increasing and likewise IVE studies are stratified by vaccine type or even vaccine brand, because it can introduce bias that should be appropriately accounted for.

The perception of different brands of a single vaccine type was less clear than the perception of different vaccine types. Most GPs in Austria, Italy and Spain perceive different brands of egg-based QIV to be clinically equivalent, whereas nearly half the GPs in the UK did not, although almost all had only a single brand of egg-based available.

Differences were observed between the four countries in how seasonal influenza vaccine is offered to individuals in risk groups that are part of the vaccine recommendations. Whilst the main approach in Austria, Italy and Spain is to offer influenza vaccine when patients present to the practice, in the UK all risk patients are contacted. In addition, GPs in the UK receive financial incentives to deliver influenza vaccinations whereas GPs in the other countries do not [25]. These factors may contribute to the higher vaccine coverage among adults ≥65y observed in the UK (72.0% in 2018–2019 [26]) compared to the other countries (53.1% in Italy [27] and 52.1% in Spain [28] in 2018–2019, 14% in Austria in 2015–2016 [29]). Whilst most GPs in Austria, Italy and Spain believe that patients who perceive themselves to be a higher risk or who are more health conscious are more likely to accept influenza vaccination, this was only the case for respectively half and two thirds of GPs in the UK (although sample size was small). One explanation for this could be that in the UK vaccination is the norm (with its school-based programs for children [26] and vaccine coverage in adults ≥65y is among the highest in Europe [7]). Prior research using discrete choice experiments has shown that the propensity to vaccinate increases with increasing vaccine coverage among acquaintances and in the population [30, 31]. Influenza vaccine acceptability in the UK may therefore be less susceptible to individual perception of vaccination than in other countries. Furthermore, in Austria influenza vaccination is not offered free of charge and patients usually have to buy the vaccine at the pharmacy prior to administration at the GP practice, which may constitute a

barrier to vaccine uptake [29, 32]. Accessibility has previously been shown to be an important determinant of vaccination [30, 31]. Finally, GPs emphasized the importance a patient's trust in the GP and health services and provision of adequate information on vaccination were important drivers of vaccine acceptance.

## Study limitations

In three countries, GPs were selected based on their participation in the national sentinel influenza surveillance (Austria, Italy) and/or the DRIVE IVE studies (Austria, UK), and may therefore not be representative of all GPs in the country. Few characteristics on GPs were collected. However, within each country, responding GPs came from a wide geographic distribution. The number of GPs surveyed in the UK was relatively low, which makes the results less representative for GPs in the UK in general; however it provides full representativeness of all GPs participating in the DRIVE network 2019/20. For the Italian survey, there was no active approach to solicit completion of the survey, hence the relatively low response rate was to be expected. The higher response rates in Austria and the UK were likely due to the direct personal outreach. Non-response bias cannot be excluded; for example GPs participating in the survey could be more interested in influenza which may translate in differences in how they approach the seasonal influenza vaccination campaigns. GPs were asked about any bias in vaccine choice they may have; if there is hesitancy to admit that the true impact of patient characteristics on vaccine choice may be larger. In all four surveys, the question on whether the GP's choice of vaccine type was directed by vaccine availability could have been more elaborate, to better understand if fluctuating stock plays a role.

## Conclusion

To our knowledge, this is the first work exploring choice of influenza vaccine type by GPs in multiple European countries. For risks groups for which more than one vaccine type is recommended, the main factor determining which vaccine type an individual patient receives is vaccine availability, mainly because only one vaccine type is available to the GPs for the campaign or alternatively due to fluctuating stock throughout the season. In addition, the role of vaccinee characteristics in determining the vaccine type used for an individual played an important role, especially in the age groups ≥65y. The most frequently mentioned characteristics that may influence choice of vaccine type were older age and the presence of multiple comorbidities. Although many factors can influence IVE estimates in multi-site studies, knowing that a non-patient related factor usually determines the vaccine type a patient receives, in settings where more than one vaccine type is recommended for a risk group, simplifies any comparisons between vaccine types in type-specific IVE studies.

## Supporting information

**S1 File. Surveys.**
(PDF)

**S2 File. Raw data.**
(XLSX)

## Acknowledgments

The authors would like to acknowledge all the GPs that participated in the survey, and Ulrike Baum (German translation), Javier Diez-Domingo (distribution in Valencia), Uy Hoang (data collection England), Miriam Levi (review), and Ana Goios (design Fig 1).

## Author Contributions

**Conceptualization:** Anke L. Stuurman.

**Formal analysis:** Anke L. Stuurman, Sara Ciampini, Cintia Muñoz-Quiles.

**Investigation:** Antonino Bella, Harshana Liyanage, Monika Redlberger-Fritz, Roberto Bonaiuti.

**Methodology:** Anke L. Stuurman, Sara Ciampini, Alfredo Vannacci, Antonino Bella, Caterina Rizzo, Cintia Muñoz-Quiles, Elisabetta Pandolfi, Harshana Liyanage, Mendel Haag, Monika Redlberger-Fritz, Roberto Bonaiuti, Philippe Beutels.

**Project administration:** Antonino Bella, Harshana Liyanage, Monika Redlberger-Fritz, Philippe Beutels.

**Resources:** Roberto Bonaiuti.

**Validation:** Anke L. Stuurman, Sara Ciampini, Cintia Muñoz-Quiles.

**Visualization:** Anke L. Stuurman.

**Writing – original draft:** Anke L. Stuurman, Sara Ciampini, Cintia Muñoz-Quiles, Mendel Haag.

**Writing – review & editing:** Alfredo Vannacci, Antonino Bella, Caterina Rizzo, Elisabetta Pandolfi, Harshana Liyanage, Monika Redlberger-Fritz, Roberto Bonaiuti, Philippe Beutels.

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
