## [Decision Letter · Decision Letter 0]

15 Mar 2021

PONE-D-21-00741

Factors driving choices between types and brands of influenza vaccines in general practice in Austria, Italy, Spain and the UK

PLOS ONE

Dear Dr. Stuurman,

Thank you for submitting your manuscript to PLOS ONE. After careful consideration, we feel that it has merit but does not fully meet PLOS ONE’s publication criteria as it currently stands. Therefore, we invite you to submit a revised version of the manuscript that addresses the points raised during the review process.

We look forward to receiving your revised manuscript.

Kind regards,

Holly Seale

Academic Editor

PLOS ONE

Journal Requirements

2. To comply with PLOS ONE submission guidelines, in your Methods section, please provide additional information regarding your statistical analyses. In addition, please report your p-values to support your claims. For more information on PLOS ONE's expectations for statistical reporting, please see https://journals.plos.org/plosone/s/submission-guidelines.#loc-statistical-reporting

3. In your manuscript you state that "surveys are available in the Supplementary Material." However, we have not received these. Please upload the surveys as supplementary material.

„I have read the journal's policy and the authors of this manuscript have the following competing interests:Phillipe Beutels reports grants from Innovative Medicines Initiative of the European Commission and attended meetings of an advisory board on economic evaluations of vaccines convened by Pfizer in 2019, outside the submitted work. Mendel Haag is employed by Seqirus BV.”

4a. Please confirm that this does not alter your adherence to all PLOS ONE policies on sharing data and materials, by including the following statement: "This does not alter our adherence to  PLOS ONE policies on sharing data and materials.” (as detailed online in our guide for authors http://journals.plos.org/plosone/s/competing-interests).  If there are restrictions on sharing of data and/or materials, please state these. Please note that we cannot proceed with consideration of your article until this information has been declared.

4b. We note that one or more of the authors are employed by a commercial company: P95 Epidemiology & Pharmacovigilance; Seqirus BV

(2) Please also provide an updated Competing Interests Statement declaring this commercial affiliation along with any other relevant declarations relating to employment, consultancy, patents, products in development, or marketed products, etc.  

Reviewers' comments:

Reviewer's Responses to Questions

**Comments to the Author**

1. Is the manuscript technically sound, and do the data support the conclusions?

Reviewer #1: Yes

Reviewer #2: Yes

2. Has the statistical analysis been performed appropriately and rigorously? 

Reviewer #1: Yes

Reviewer #2: Yes

3. Have the authors made all data underlying the findings in their manuscript fully available?

Reviewer #1: Yes

Reviewer #2: Yes

4. Is the manuscript presented in an intelligible fashion and written in standard English?

Reviewer #1: Yes

Reviewer #2: Yes

5. Review Comments to the Author

Reviewer #1: This paper presents the descriptive results of a survey conducted among GP's in four European countries on how they promote influenza vaccination in their clinics and how they choose which influenza vaccine to administer to patients. These findings have implications for vaccine effectiveness estimates, as well as offer insight into vaccine selection when multiple influenza vaccine options are available. The statistical methods used are basic, but I believe they are appropriate.

Specific comments below:

Introduction: nicely written

Methods:

- Lines 130-133: The methods do not explain how open-ended responses were analysed and categorised into themes. Please describe what methods were used. Were themes defined a priori?

- It would be interesting to know how the survey questions were developed. Were they derived from previous research or designed specifically for this study? If the latter, was any theoretical framework used in their development?

Discussion:

-The first paragraph is somewhat repetitive at times. For example, lines 291-296 and lines 313-315 present similar information, as well as lines 335-337 and 304-307 . I suggest rewriting the paragraph to make it somewhat more concise.

-Lines 378-380: As you mention, response rates in Italy were quite low (14%). It would be helpful to have some more discussion on how this may have biased results.

Reviewer #2: Review comments for manuscript: Factors driving choices between types and brands of influenza vaccines in general practice in Austria, Italy, Spain, and the UK

As an important component to advancing evaluation of effectiveness of influenza vaccines, Anke L et al conducted a descriptive study to describe the factors that influence the choice in use of influenza vaccine types and brands among general practitioners and risk category vaccinees in four European Countries.

Annually, several influenza vaccines are recommended in Europe. Factors influencing the use of these vaccines are many and can vary by health providers as well as by those getting vaccinated. The authors conducted a study to better understand these factors in 4 European countries. The research question/s proposed by the authors are of public health importance.

Overall, the manuscript is well written. It is easy to read.

Specific comments:

Methods: The research methods are clear and straightforward. However, the inclusion of Spain-Valencia is somewhat confusing given it was not participating in the DRIVE project. The choice of this site requires a stronger rationale (Line 94-98). Or was it convenient?

Information on how the reliability of the survey tool used was assessed would be beneficial to add. Similarly, regarding the online survey tool, information has not been provided on whether participation could occur more than once?

Details on the sample size that was sufficient to answer the study questions are missing.

Results

The results are well presented.

Table 2, lines 145-150: the numbers in parenthesis are not clearly explained and this is somewhat confusing.

Some results are presented as absolute numbers only; other results are presented as absolute numbers (proportions). It is best to be consistent.

A comment on huge difference in the response rates across countries would be useful.

Discussion

The discussion is well written. Limitations and areas for future research are well captured in the manuscript. Is there any bias that could have made the results less reliable?

The manuscript is well written with good scientific and public health applications. The authors have done a good work.

6. PLOS authors have the option to publish the peer review history of their article (what does this mean?). If published, this will include your full peer review and any attached files.

Reviewer #1: No

Reviewer #2: **Yes: **Benjamin Kagina

---

## [Author Response · Author response to Decision Letter 0]

12 May 2021

Dear Editor,

We have today uploaded:

Rebuttal letter

Tracked-changed manuscript

Manuscript

Suppl.material 1

Suppl. material 2

Cover letter with Funding Statement and Competing Interests.

By that, we believe we have addressed all your comments.

The Figure we submitted last time shall stay also here in the re-submission.

Kind regards

---

## [Editor Report · Decision Letter 1]

24 May 2021

Factors driving choices between types and brands of influenza vaccines in general practice in Austria, Italy, Spain and the UK

PONE-D-21-00741R1

Dear Dr. Stuurman,

We’re pleased to inform you that your manuscript has been judged scientifically suitable for publication and will be formally accepted for publication once it meets all outstanding technical requirements.

Kind regards,

Holly Seale

Academic Editor

PLOS ONE
---

## [Editor Report · Acceptance letter]

7 Jun 2021

PONE-D-21-00741R1 

Factors driving choices between types and brands of influenza vaccines in general practice in Austria, Italy, Spain and the UK 

Dear Dr. Stuurman:

I'm pleased to inform you that your manuscript has been deemed suitable for publication in PLOS ONE. Congratulations! Your manuscript is now with our production department. 

Kind regards, 

on behalf of

Dr. Holly Seale 

Academic Editor

PLOS ONE